# Development and Production of Artificial Test Swarf to Examine Wear Behavior of Running Engine Components—Geometrically Derived Designs

Patrick Brag [1,*], Volker Piotter [2], Klaus Plewa [2], Alexander Klein [2], Mirko Herzfeldt [3] and Sascha Umbach [4]

1    Department of Ultraclean Technology and Micromanufacturing, Fraunhofer Institute for Manufacturing Engineering and Automation IPA, Nobelstrasse 12, 70569 Stuttgart, Germany
2    Institute for Applied Materials (IAM-WK), Karlsruhe Institute of Technology (KIT), Hermann-von Helmholtz-Platz 1, 76344 Eggenstein-Leopoldshafen, Germany
3    AuE Kassel GmbH, Heinrich-Hertz-Str. 52, 34123 Kassel, Germany
4    Department of Machine Elements and Engineering Design (iaf), University of Kassel, Mönchebergstr. 7, 34125 Kassel, Germany
*    Correspondence: patrick.brag@ipa.fraunhofer.de; Tel.: +49-711-970-1104

**Abstract:** Subtractive manufacturing processes are usually accompanied by the occurrence of tiny flakes and swarf, which later on cause severe wear and damage, especially in moving components such as rolling or sliding bearings, pistons, etc. However, up until now, such detrimental effects have hardly been investigated. One reason is the lack of a definition of a typical design of debris particle. Therefore, the main goal of the project described in this paper was to elaborate a draft that defines standardized test particles. It had to be evaluated whether test particles could be adequately reproduced and whether they would reveal significant damage potential. Taking into account future mass fabrication, Micro Powder Injection Molding (MicroPIM) was chosen as a production method. Five different 3D designs of geometrically defined test particles were developed. The maximum size of each design was 1167 mm in green state; however, all samples shrank in size during sintering. Specially tailored feedstocks containing 42CrMo4 steel powders were used and the related molding, debinding and sintering procedures were developed. All particle geometries and related mold inserts were developed using a commercial software routine for the layout of runner systems, gate locations and ejector positions. The damage potential of the test particles was evaluated based on trials using journal bearing and shift valve test rigs. Although only a moderate degree of damage potential could be ascertained up until now, it can be expected that the artificial swarf will enable standardized wear test procedures to be developed.

**Keywords:** MicroPIM; particle; shape characterization; damage; tribology; bearing

## 1. Introduction

Manufacturing debris, such as solid particulate contamination originating from machining processes, can cause severe damage in engines, transmissions and other moving components in engineering aggregates. For example, in the years 2017 to 2019, General Motors USA was obliged to recall 208,546 Chevrolet Silverado 2500/3500 and 122,728 GMC Sierra 2500/3500 models for safety reasons. Contamination on the block-heater sealing could potentially cause coolant leaks and culminate in fire [1]. Similar safety issues have been reported for airbags [2], steering systems [3] and drive batteries [4]. Endeavors to characterize the nature and occurrence of contaminants in transmissions can be traced back to the year 1990 [5]. The search for an ideally natural contamination mixture and production thereof was driven by the lack of alternatives. At best, tribological studies were carried out with Arizona test dust [6] or aluminum oxide test dust [7]. Nowadays, computational fluid dynamics (CFD) is used to simulate tribological behavior. First, with spherical particle

representations [8], later with non-spherical but regular composed geometries [9]. With the enduring progress in 3D image processing [10], discrete element method (DEM) is no longer limited to such abstract objects. A digital twin of nearly every particle can be generated by micro computed tomography and deployed for CFD simulation [11,12]. The alteration of the surface microstructure as a result of particle-wall collisions is still part of current studies [13]. So far, however, a comprehensive scientific investigation based on real experimental results has not been carried out. One reason is the lack of standardized micro-sized test particles needed in order to make an actual reliable quantitative analysis of debris-wear correlations. The final goal of this collaborative research project was to create a draft norm specification for standardized test particles. Several tasks had to be solved, the most important being to develop artificial particle designs, to build up at least one mass production route, to evaluate whether this process runs sufficiently reproducibly, and to ascertain whether the test particles reveal significant damage potential. However, some of these aspects also occur in macroscopic dimensions: Whereas the choice of micro metal injection molding (MicroMIM) was mandatory for this project, particular challenges like discontinuous thicknesses and non-regular geometries are important topics for larger parts, too. Further on, micro specific features such as variothermal temperization may help in the micro as well as in the macroscopic world.

## 2. Design Development

As natural debris of subtractive manufacturing processes, real swarf occur in a randomly distributed diversity of size, shape, thickness, etc. Therefore, two principally different design approaches were followed:

- Designs derived from geometrically consistent bodies, hereinafter called abstracted particles;
- Designs derived from an accidentally chosen real swarf, hereinafter called native test swarf/particles.

Targeting the first approach, five different designs were defined, geometrically and by name, as illustrated in Figure 1.

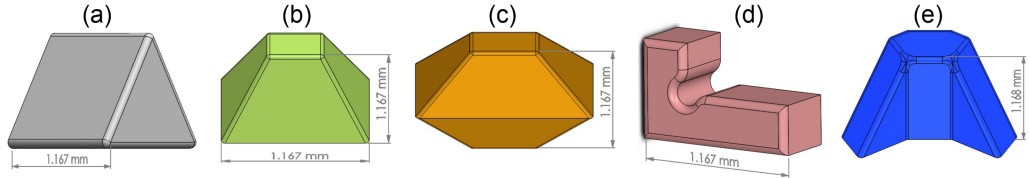

**Figure 1.** Geometrically defined test particles: (**a**) Triangular prism. (**b**) Truncated pyramid. (**c**) Octahedron. (**d**) L-angle. (**e**) Abstract swarf.

The second approach, i.e., design development based on real swarf measured by 3D technique, will be the subject of a further publication.

## 3. Modification of the Micro Powder Injection Molding Process

The manufacturing process of micro powder injection molding (MicroPIM) was chosen. Although originally developed for Micro System Technology (MST) applications, this technique offers the possibilities of using the same type of steel as that of real swarf, and also enables test particles to be mass produced cost-effectively. Comprehensive descriptions of the basic macroscopic PIM fabrication technology can be found in e.g., [14,15], whereas introductions to the micro-specific variant are given in e.g., [16–19]. Some aspects of the MicroPIM process had to be modified, i.e., feedstock development, tool layout, mold insert layout, and parameter sets for injection molding, debinding and sintering. The first step in the new process line was the composition of the feedstock. Low-alloyed 42CrMo4 (1.7225) heat treatable steel was chosen as a powder since this material is widely used in combustion engines. The particular fraction utilized was provided by Sandvik Osprey Ltd. (Sandviken, Sweden) It had a D10 of 2.9 µm and a D90 of 5.7 µm as determined by laser interferometry.

The BET surface was determined to be 0.25 m$^2$/g. As a binder, the so-called GoMikro system was chosen, which has been proven at KIT to be highly suitable for extremely small dimensions. It usually consisted of paraffin wax, polyethylene and stearic acid in a mixing ratio of 50/45/5 vol%. Following initial trials, the powder–binder ratio was set at 63 vol%, in line with the processing demands of low viscosity and high green strength. The mixture was compounded in a measurement kneader until a constant mixing torque was reached. Simulating the filling and cooling phase can be a quite useful aid to avoid basic mistakes in the tool and mold insert layout. It has been often deployed for powder injection molding tasks, not at least to predict areas of powder–binder segregation caused by extremely high shear rates, such as those occurring in micro-sized cavities [20]. Therefore, this method was applied to redesign one of KIT's experimental tools for manufacturing the test particle green bodies. The runner system consisted of a thick circle distribution cavity which had to guide the feedstock resin into four conical runners, each one generating a micro-sized part cavity. The lengths and thicknesses of the runner system were adjusted in such a way to prevent maximum shear stresses from being largely exceeded 40,000.00 s$^{-1}$ at any point, which is an empirical threshold ascertained from various previous experiments (Figure 2). A moderate shear rate is of paramount importance in order to prevent severe powder–binder segregation. The latter is probably the most important reason for inhomogeneous debinding rates that cause severe failures during sintering. Simulating mold filling process helps to avoid these difficulties by optimizing the runner and cavity design before starting the actual mold production.

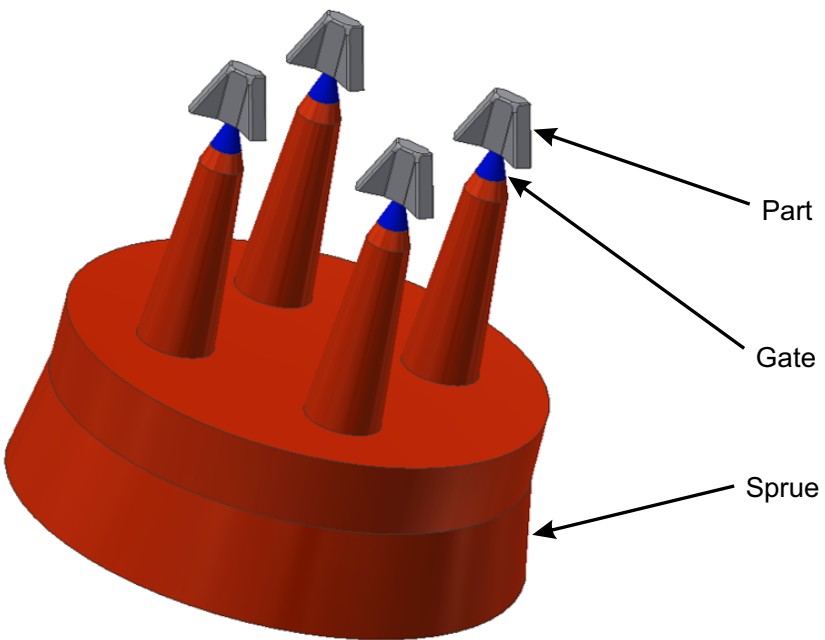

**Figure 2.** Schematic drawing of the runner system supplying four cavities, here abstract swarf.

The next task was to design the micro cavities themselves. Again, the maximum shear rates had to be kept as low as possible, which was a challenging demand as the gate thickness could not be larger than 0.28 mm, i.e., no larger than the width of the thinnest test part geometry (L-shape). However, even by maximizing the gate diameter, shear rates of up to 44,000.00 s$^{-1}$ may occur, i.e., a level which is barely tolerable. Shear rates could only be further reduced by adjusting the injection parameters (velocity, pressure) and not by modifying the tool design, as an increased gate diameter would exceed the dimensions of the micro cavities themselves. An example of the various simulation results obtained during the mold layout phase is shown in Figure 3.

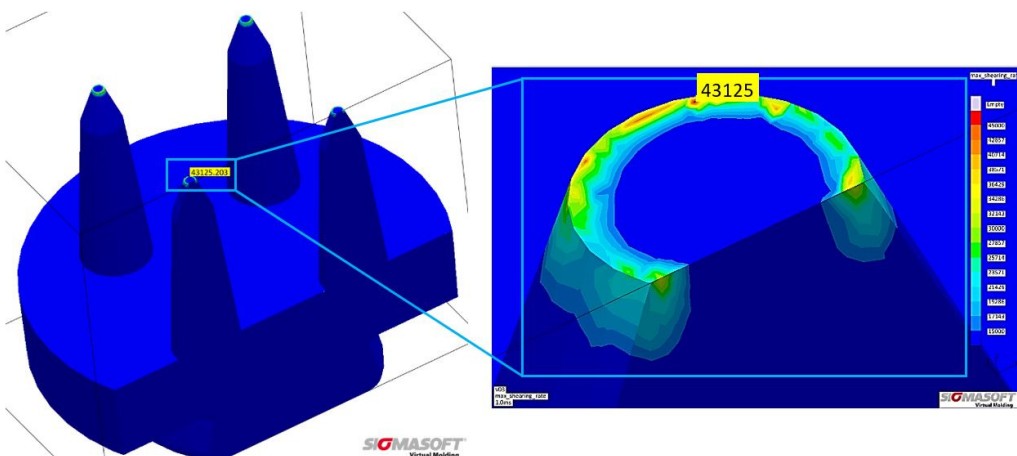

**Figure 3.** Simulated shear rate profile occurring at the gate (diameter 0.28 mm).

After completing the cavity simulations, the required mold inserts were manufactured using a combination of micro milling and micro discharge machining. Each test particle design had been allocated to a so-called stamp, as shown in Figure 4. This way, all variants of test particles could be produced by changing the particular stamps while using the same basic molding tool.

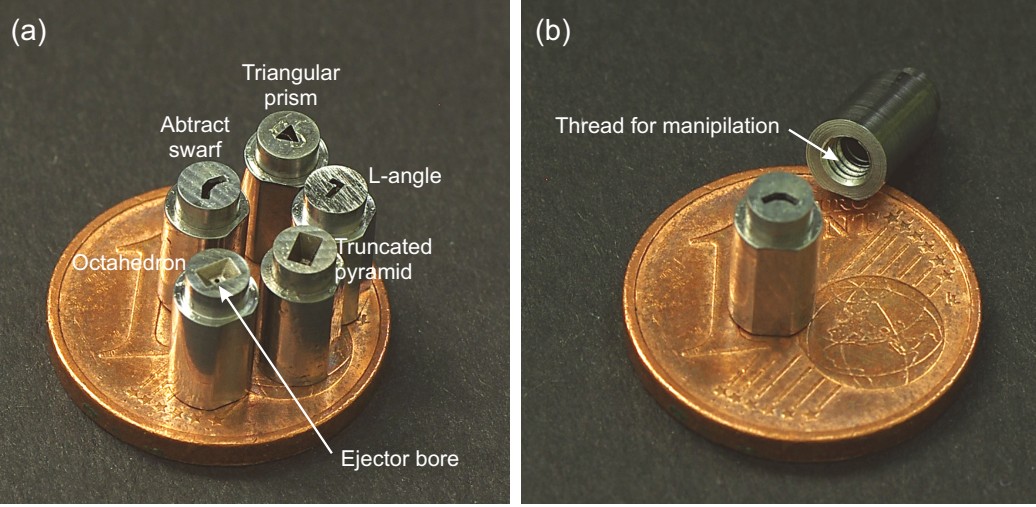

**Figure 4.** Stamps: (**a**) Each one carrying a cavity of one of the five test particle designs on the front side. (**b**) Screw thread on the back side for handling and fixation in the molding tool.

For the injection molding process, a Battenfeld Microsystem 50 machine specially tailored for molding micro-sized components was used. A particular feature was the application of the so-called variothermal temperization technique, i.e., tool temperatures were heated up prior to injection to keep the feedstock on an easy flowing level and were cooled down once the cavities were filled to achieve green bodies strong enough to be removed safely from the mold. In the course of the prior running-in period, the following parameters were identified as being the most suitable (Table 1).

The debinding process was performed in a two-step manner, i.e., a prior dissolving step to remove the low molecular components (wax, stearic) was followed by thermal extraction of the thermoplastic component (PE). The first step took place at 30 °C in n-hexane for a period of 3 h. Thermal debinding was incorporated in the heat treatment process, and was thus immediately succeeded by the sintering procedure. The latter was adapted to micro dimensions, i.e., the parameters given by the powder supplier/found in literature had to be varied slightly. The best values were determined by analysis using optical microscopy with the aim of achieving a

microstructure as fine-grained as possible. Finally, compared to industrial sintering procedures for 42CrMo4 parts of macroscopic size [21], the holding time was reduced while increasing the heating rates. Both modifications were necessary due to the small dimensions of the test particles. The entire parameter set for the heat treatment step is shown in Table 2. Some images of the sintered test particles can be seen in Figure 5.

**Table 1.** Injection molding parameters used.

| Parameter | Value |
| --- | --- |
| Injection unit temperatures | 160/165/160 °C |
| Tool temperature, nozzle side, injection | 95 °C |
| Tool temperature, nozzle side, demolding | 25 °C |
| Tool temperatures, ejector side, injection | 60 °C |
| Tool temperature, ejector side, demolding | 20 °C |
| Injection velocity | 125 mm/s |
| Injection pressure | 1350 bar |
| Holding pressure | 620 bar |

**Table 2.** Parameters applied for the combined thermal debinding and sintering procedure.

| Process Steps | Start Temperature [°C] | Heating Rate [K/min] | End Temperature [°C] | Holding Time [min] | Atmosphere |
| --- | --- | --- | --- | --- | --- |
| Debinding | RT | 1.5 | 600 | 30 | $N_2$ |
| Sintering | 600 | 15 | 1290 | 5 | $N_2$ |
| Cooling I | 1290 | −15 | 700 | - | $N_2$ |
| Cooling II | 700 | <−15 | RT | - | $N_2$ |

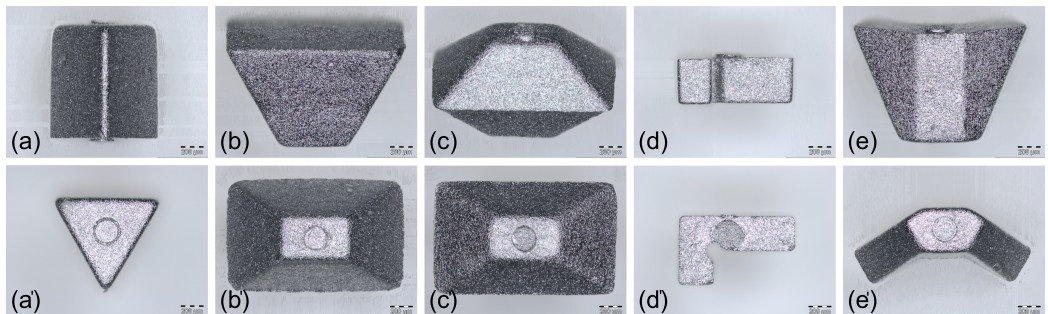

**Figure 5.** Finally sintered geometrically defined test particles: (**a**,**a′**) Triangular prism. (**b**,**b′**) Truncated pyramid. (**c**,**c′**) Octahedron. (**d**,**d′**) L-angle. (**e**,**e′**) Abstract swarf.

## 4. Results

The final samples were characterized directly after sintering. As is common in PIM, the achieved densities were the first material data to be determined. Measurements by He-pycnometry revealed a medium density of 7.702 g/cm$^3$ with a standard deviation of 0.0059 g/cm$^3$. Compared to a density of 7.72 g/cm$^3$ for bulk 42CrMo4 steel samples [22], these values correspond to 99.7% of the theoretical density and are thus quite a good result. Due to the minimum mass requirement of the He-pycnometry measurement method, this value is an average result of more than 10 parts. A critical point in MicroPIM concerns the grain size of the microstructure after sintering because it affects the isotropy of the mechanical properties. Microscopic analysis of cross-cuts of the L-shaped samples revealed microstructures as shown in Figure 6. The samples were prepared in a first step by grinding with 1000 grit paper followed by polishing rags with 9 µm and 3 µm. The second step was an etching process with 2% nital solution for 15 s.

Despite the faster cooling rates (see Table 2), which usually result in a fine grain, only a low amount of relatively small-sized grains could be determined. Such rough microstructures—in relation to the dimensions of the parts—often result in anisotropic mechanical behavior of the entire sample. However, as the task of the artificial swarf was

not to reach highest strength but instead to reveal a damage potential comparable to that of real swarf, the low number of grains per area was considered as just acceptable.

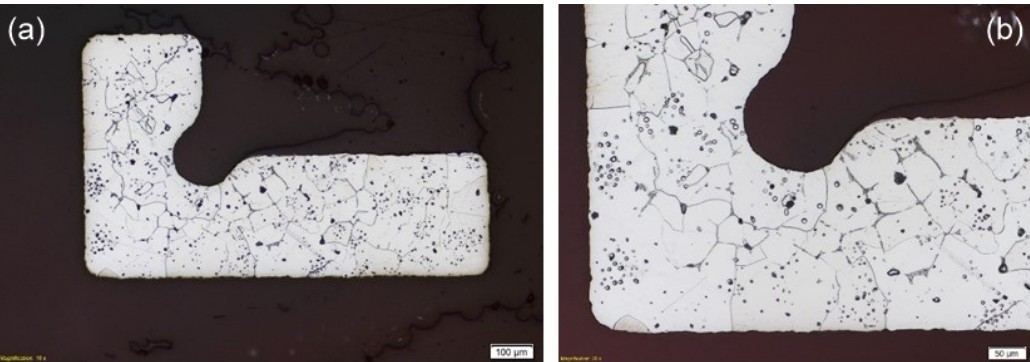

**Figure 6.** Cross-cut of an L-angle sample showing the microstructure as obtained after sintering: (**a**) in full size and (**b**) enlarged.

The next task was to determine the dimensional accuracy of the samples and the repeatability of the MicroPIM process. To this end, 33 samples each of L-shape and abstract swarf were characterized by micro computed tomography (micro CT). The dimensions measured were the length and thickness [23]. The artificial test swarf was scanned with a resolution of 1.73 μm per voxel. Using the best fit algorithm of VGSTUDIO MAX 3.0.5 (Volume Graphics GmbH, 69115 Heidelberg, Germany), each scanned object was aligned with the underlying CAD model, which had been used before to machine the tools. A nominal/actual comparison was made to calculate the deviation of each test particle from its CAD model; see Figure 7.

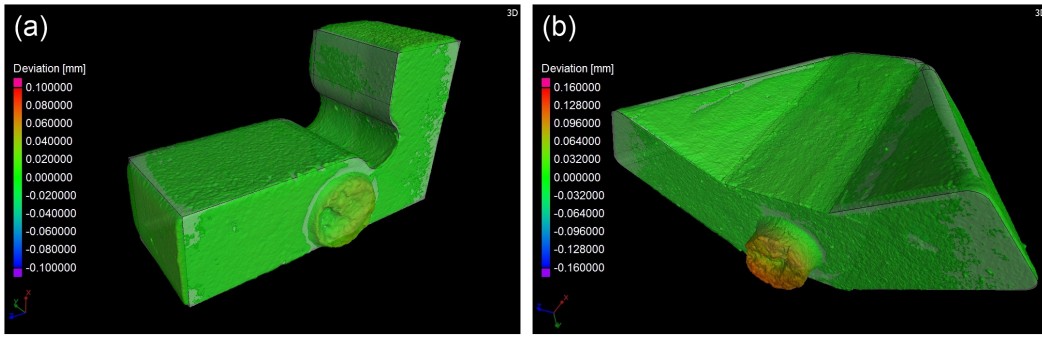

**Figure 7.** Nominal/actual comparison: (**a**) L-angle. (**b**) Abstract swarf.

The histogram plot of a representative L-shaped particle illustrates the achieved accuracy of the MicroPIM process. Figure 8 shows an excess of 11.21% for a threshold of ±10 μm. This excess can be mostly attributed to the runner overhang and ejector impression.

A corresponding evaluation was made from various histogram plots of the abstract swarf; see Figure 9. Although the minimum and maximum deviation is higher compared to those of the L-shaped particle, the tolerance excess is lower with 5.83%, which indicates a better body accuracy.

The following box plots for L-shaped and abstract swarf were composed on the basis of absolute values for Quantile 1, 2 and 3 and actual values for minimum/maximum deviations to better depict the natural spread; see Figures 10 and 11. An illustration of the absolute cumulative values would only lead to a reduction in information.

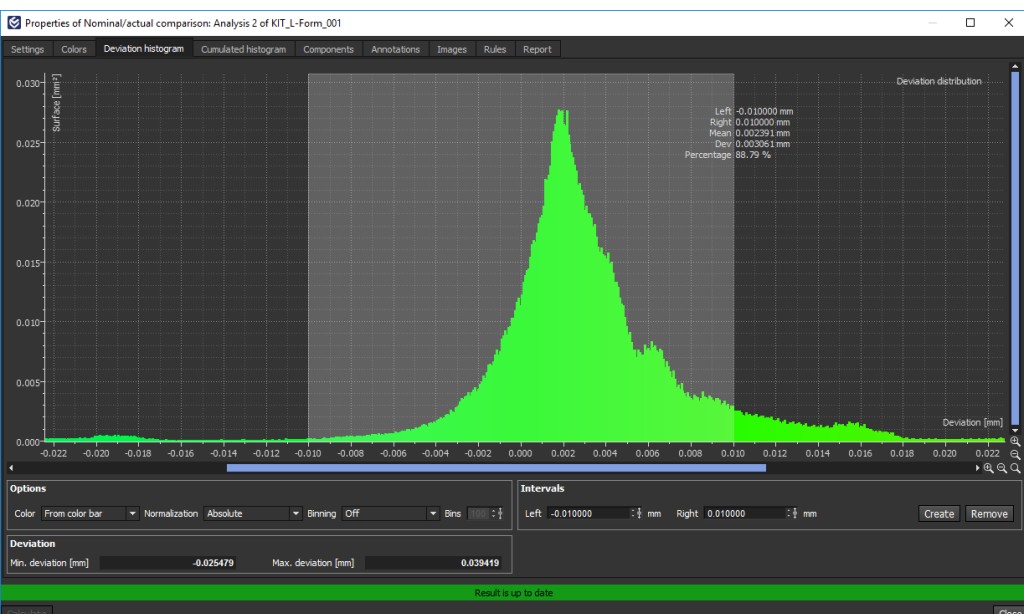

**Figure 8.** Deviation histogram of analyzed L-angle particle No. 1.

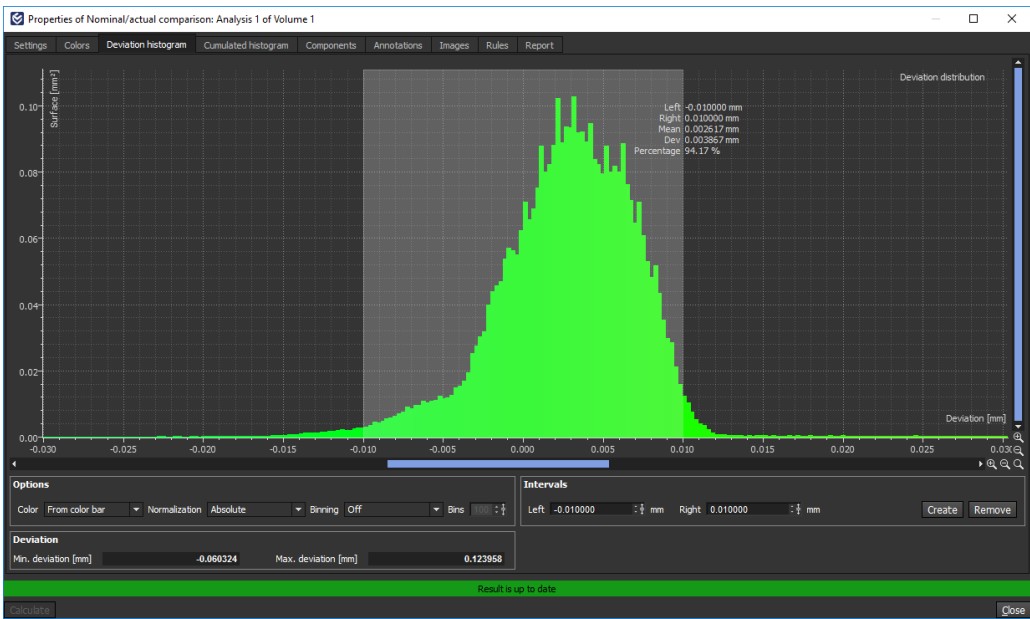

**Figure 9.** Deviation histogram of analyzed abstract swarf No. 4.

Unfortunately, due to some surface defects in the scans, not all abstract swarf scans could be used for the box plot; see Figure 11. The remaining 26 results are, however, highly consistent with the results of the L-shaped particle, with good body accuracy and higher deviations at the runner and ejector contact points.

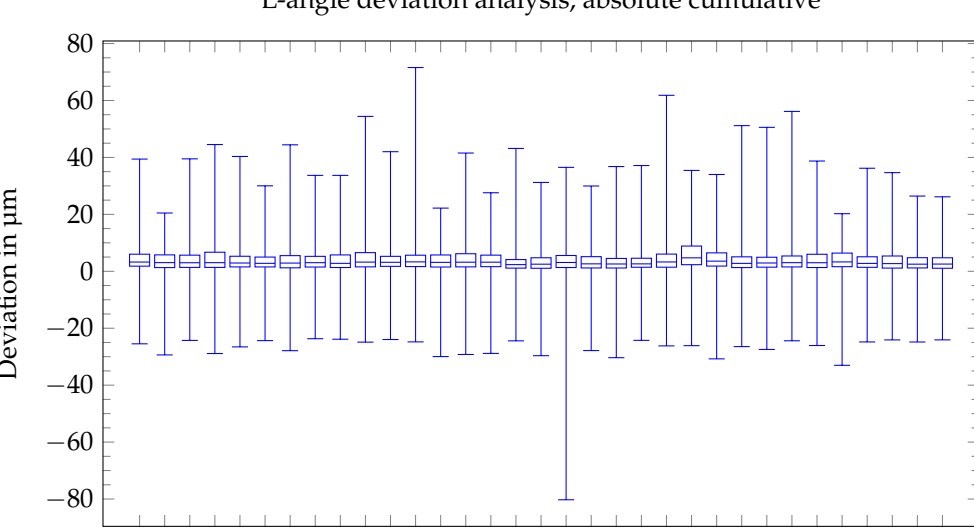

**Figure 10.** Box plots of the deviation analysis for the L-angle particle.

**Figure 11.** Box plots of the deviation analysis for the abstract swarf particle.

## 5. Investigation of the Artificial Swarf

For the particle compatibility tests on the artificial swarf, a plain bearing test bench already available at the University of Kassel's Institute for Machine Elements and Tribology was used [24]. It had formerly been utilized in numerous projects concerned with particle compatibility tests on engine plain bearings. In particular for the BMWi joint project described in this paper, a special particle injector was developed and used at a later stage to deliver the particles to the corresponding critical point of action (Figure 12). This means that test particles could be introduced into the oil flow at any time while tests were running. By pressing the feed button, the particle was injected into the oil supply and transported by the oil to the point of action.



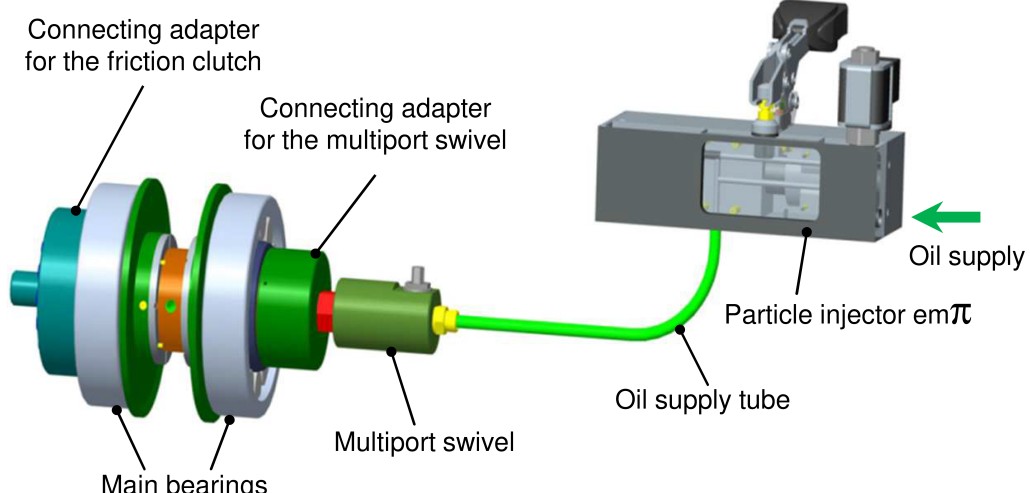

**Figure 12.** Particle injector for slide bearings.

The particle was detected at the point of action based on the principle of measuring the contact resistance between the bearing and the shaft. If the metal bearing shell was mechanically damaged and the particle made contact between the shaft and the bearing, there was an electrical short circuit in the components, which was detected at high frequency. Due to the large amount of data, these values were first discretized and then applied as a percentage as mixed friction intensity over one working cycle (720° crank angle) through summation and averaging. The procedure is shown in Figure 13.

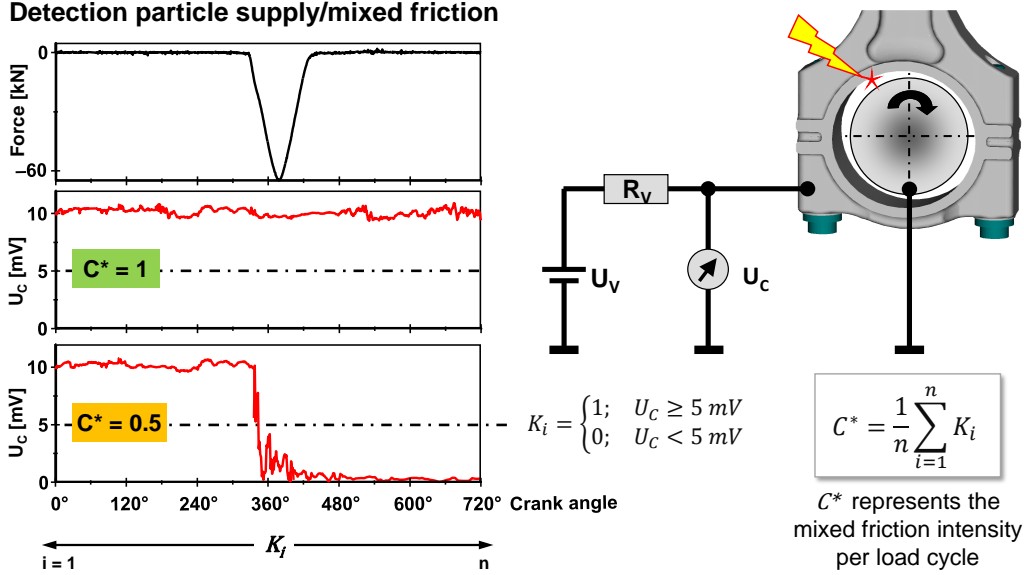

**Figure 13.** Definition of mixed friction coefficient and schematical drawing of the contact detection system.

Figure 14 shows the investigated L-shaped and octahedron-shaped particles before and after the test run. Neither type was destroyed, but merely ground smooth. The low degree of wear/damage of the particles indicates that the bearing material was not penetrated, as would have been the case if milling swarf had been used, for example. In further tests, partial break-offs from particles up to complete attrition of the particles could also be observed.

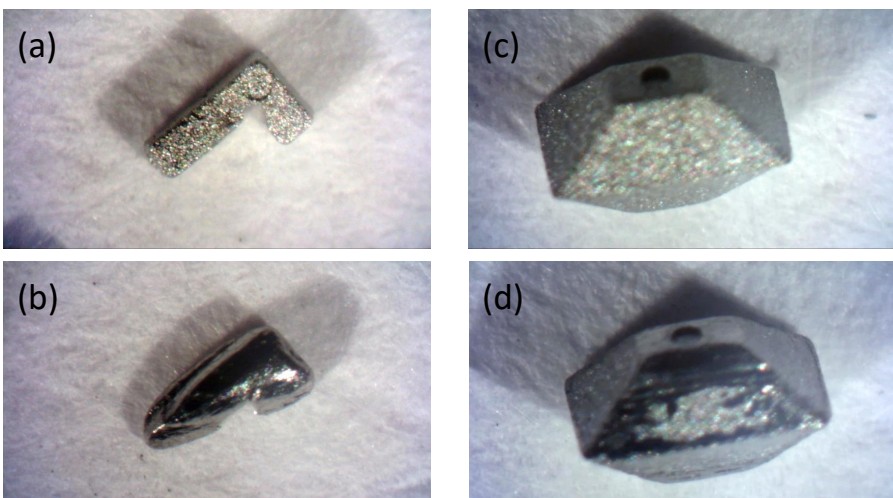

**Figure 14.** Geometrically derived particles: (**a**) L-angle before experiment. (**b**) L-angle after experiment. (**c**) Octahedron before experiment. (**d**) Octahedron after experiment.

The geometry of the particles appeared to have a major influence on penetration behavior and thus on the test result. Figure 15 shows the course of the mixed friction intensity of different particle geometries. Increased proportions of mixed friction for the L-shaped and octahedron-shaped particles, i.e., increased proportions of solid body contacts between the bearing shell and shaft, were determined. The course of mixed friction confirmed the picture of the partially damaged particles. No mixed friction components were detected for the abstract particle shape (Figure 15), top measurement plot and the pyramid. Neither bearing shell showed no signs of damage after the test. The remaining three particle shapes could damage the bearings with circumferential surface scoring. In addition to the typical run-in wear, a circumferential groove in the bearing material was seen on completion of the test. Compared to the usual damage caused by particle entry, damage was only minimal. The damage pattern corresponds to the measured proportions of mixed friction and the wear pattern of the particles (see Figure 15).

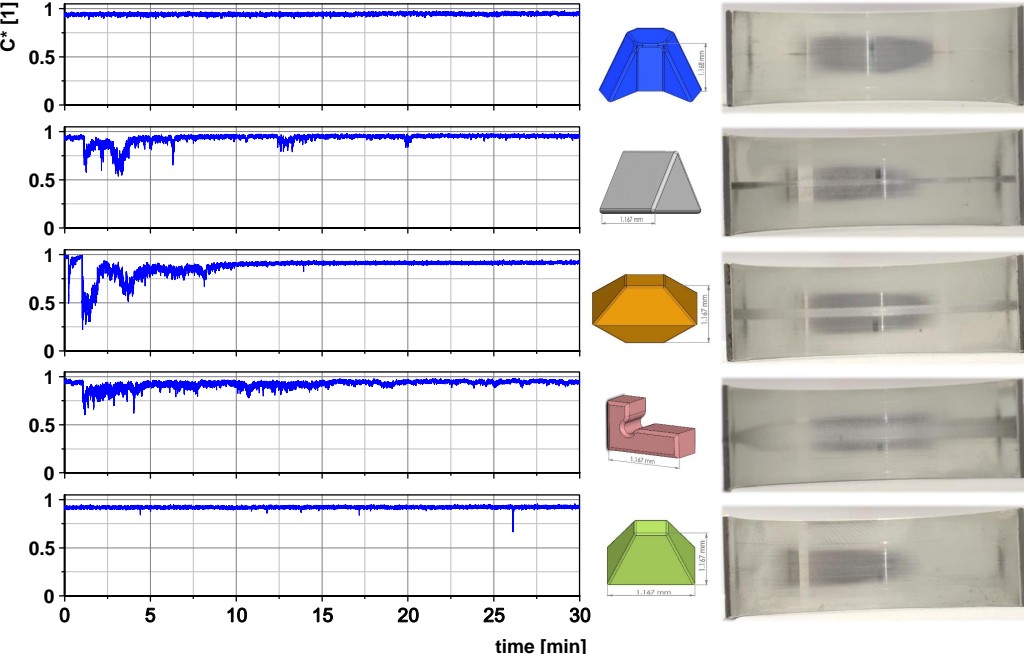

**Figure 15.** Mixed friction behavior of different particle shapes.

## 6. Conclusions and Outlook

Based on the results presented, it can be concluded that the behavior of the injection-molded particles under load is significantly different to that of particles produced by cutting, despite being made of the same material. The damage patterns of the injection-molded particles are generally less pronounced and did not lead to bearing failure in any of the tests, i.e., the degree of damage to the bearing shell is not critical for further operation. However, such low damage potential would hardly be sufficient to use the particles in the aspired wear tests. To improve this significantly, two approaches look promising:

- Utilization of a much harder material, such as hardened steel or even ceramics;
- Development of sharper designs derived from real swarf, as Section 2.

The disadvantage of the first approach is that a novel material would be used in the tests that is not implemented in industry in practice, thus giving rise to further questions concerning aspects such as brittleness and residual debris. By comparison, the second approach would result in test particle designs nearer to those encountered in reality—an additional benefit. Consequently, the second approach was chosen and will be the topic of a further publication. It has to be emphasized that this approach also incorporates the challenge of real free-formed geometries, i.e., discontinuous dimensions in all spatial directions. This includes major thickness fluctuations, thus violating basic rules of PIM part layout. Typical micro molding features like variothermal temperization may help to solve the problems caused by inhomogeneous filling, debinding and sintering conditions.

To improve dimensional accuracy (see section on "Results"), two paths are commonly taken in PIM production:

- Slightly higher content of binder in the feedstocks, which increases sintering shrinkage and thus results in smaller PIM parts;
- Modified tool designs, i.e., altered dimensions of the micro-sized cavities.

Since the first approach is clearly the most cost-effective, it is thus favored for further experiments.

**Author Contributions:** Conceptualization, V.P. and P.B.; methodology, V.P., K.P., P.B. and S.U.; validation, V.P., K.P., P.B. and M.H.; formal analysis, V.P., P.B. and S.U.; investigation, V.P., K.P., A.K., P.B., M.H. and S.U.; resources, V.P., P.B. and S.U.; data curation, V.P., P.B. and S.U.; writing—original draft preparation, V.P.; writing—review and editing, V.P., P.B. and S.U.; visualization, P.B., V.P. and S.U.; supervision, V.P. and P.B.; project administration, P.B.; funding acquisition, P.B. All authors have read and agreed to the published version of the manuscript.

**Funding:** This research was funded by the Federal Ministry for Economic Affairs and Climate Action (German: Bundesministerium für Wirtschaft und Klimaschutz; short: BMWK), formerly: Federal Ministry for Economic Affairs and Energy (German: Bundesministerium für Wirtschaft und Energie; short: BMWi) grant numbers 03FS15021, 03FS15022, 03FS15023 and 03FS15024.

**Institutional Review Board Statement:** Not applicable.

**Informed Consent Statement:** Not applicable.

**Data Availability Statement:** Not applicable.

**Acknowledgments:** The research work described above was carried out in the frame of the collaborative research project *Standardization of test particles by test-supported approximation of the damage behavior of milled chips (test particles)* The authors gratefully acknowledge the financial support provided by the German Federal Ministry for Economic Affairs and Climate Action (BMWK, formerly BMWi) and thank all contributing colleagues and external partners for their always helpful cooperation.

**Conflicts of Interest:** The authors declare no conflict of interest.

## Abbreviations

The following abbreviations are used in this manuscript:

| | |
|---|---|
| BET | Brunauer, Emmet, Teller |
| CAD | Computer Aided Design |
| CFD | Computational Fluid Dynamics |
| DEM | Discrete Element Method |
| Micro-CT | Micro Computer Tomography |
| MicroMIM | Micro Metal Injection Molding |
| MicroPIM | Micro Powder Injection Molding |
| MST | Micro System Technology |
| PE | Polyethylene |
| PIM | Powder Injection Molding |
| RT | Room temperature |

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
