# Peer review of "Development and Production of Artificial Test Swarf to Examine Wear Behavior of Running Engine Components—Geometrically Derived Designs"

_jmmp, doi:10.3390/jmmp6050100_

Round 1

Reviewer 1 Report

The work is generally interesting, but it not clear how the microMIM work will help with much larger parts (which can accept larger particles). The Introduction could be modified to emphasize this point.

The discussion of the results should be expanded to emphasize critical points.

There are some issues with word usage.  

Author Response

A comprehensive spell and language check was done by a native speaker.

The chapters 1, 3, 4 and 6 has been slightly extended and revised.

The objective of this work was to create tiny, but regular defined test particles for real experiments, that represent the dimensions of natural contaminants, found in particle induced damages. The challenge of the process development was to scale down the manufacturing process for the targetted application of the test particles. A need of even smaller test particles is anticipated for the future. The authors see no benefit in scaling up the process. The downsizing trend in automobile industry affetcs the required size of the test particles as well.

The detailed point-by-point responses can be found in the attachment.

Reviewer 2 Report

In this paper, five different 3D designs of geometrically defined test particles were prepared by using MicroPIM method. And the related molding, debinding, and sintering procedures were developed. The damage potential of test particles was evaluated based on trials using journal bearing and shift valve test rigs. Although the authors have done a relatively comprehensive work, there are still many details to be further improved.

1. In Section 3, the authors present the result of the Simulated shear rate profile occurring at the gate, but there is a lack of simulation process and discussion. What is the significance of the simulation results for the subsequent manufacturing process?

2. How did the author determine the sintering process parameters in Table 2?

3. How did the authors characterize the samples? what equipment was used? And how were the samples prepared? Please describe the sample characterization process in detail.

4. In Section 4, what does the author mean by "the microstructure appears relatively rough and characterized by an only small, however, barely acceptable number of grains."?

5. In Section 4, the measured value of the relative density of the sample is 99.7%, is this the average value? Or just for one sample?

6. English language and style need to be improved.

Author Response

A comprehensive spell and language check was done by a native speaker.

The chapter 1, 3, 4 and 6 has been slightly extended and revised with special respect to the annotations 1-5 in particular.

The detailed point-by-point responses can be found in the attachment.

Round 2

Reviewer 2 Report

The authors have done a good job of considerably improve their manuscript. I think this article can be accepted.